# New Insights into Leaf Physiological Responses to Ozone for Use in Crop Modelling

**DOI:** 10.3390/plants8040084

**Published:** 2019-04-01

**Authors:** Stephanie Osborne, Divya Pandey, Gina Mills, Felicity Hayes, Harry Harmens, David Gillies, Patrick Büker, Lisa Emberson

**Affiliations:** 1Centre for Ecology and Hydrology, Environment Centre Wales, Bangor LL57 2UW, UK; gmi@ceh.ac.uk (G.M.); fhay@ceh.ac.uk (F.H.); hh@ceh.ac.uk (H.H.); 2Stockholm Environment Institute, Environment Department, University of York, York YO10 5NG, UK; pandey.divyaa85@gmail.com (D.P.); david.gillies@york.ac.uk (D.G.); patrick.bueker@giz.de (P.B.); l.emberson@york.ac.uk (L.E.)

**Keywords:** ozone, air pollution, wheat, photosynthesis, leaf senescence, crop modelling

## Abstract

Estimating food production under future air pollution and climate conditions in scenario analysis depends on accurately modelling ozone (O_3_) effects on yield. This study tests several assumptions that form part of published approaches for modelling O_3_ effects on photosynthesis and leaf duration against experimental data. In 2015 and 2016, two wheat cultivars were exposed in eight hemispherical glasshouses to O_3_ ranging from 22 to 57 ppb (24 h mean), with profiles ranging from raised background to high peak treatments. The stomatal O_3_ flux (Phytotoxic Ozone Dose, POD) to leaves was simulated using a multiplicative stomatal conductance model. Leaf senescence occurred earlier as average POD increased according to a linear relationship, and the two cultivars showed very different senescence responses. Negative effects of O_3_ on photosynthesis were only observed alongside O_3_-induced leaf senescence, suggesting that O_3_ does not impair photosynthesis in un-senesced flag leaves at the realistic O_3_ concentrations applied here. Accelerated senescence is therefore likely to be the dominant O_3_ effect influencing yield in most agricultural environments. POD was better than 24 h mean concentration and AOT40 (accumulated O_3_ exceeding 40 ppb, daylight hours) at predicting physiological response to O_3_, and flux also accounted for the difference in exposure resulting from peak and high background treatments.

## 1. Introduction

The air pollutant ozone (O_3_) reduces yield in many crops including wheat, rice, and soybean [1,2]. Ozone at the ground level forms from precursor gases—chiefly NO_x_ and volatile organic compounds (VOCs)—in chemical reactions catalysed by sunlight and heat [3]. Concentrations over much of the Earth’s land surface have approximately doubled since pre-industrial times, mainly due to anthropogenic emissions from vehicles, industry, and agriculture [4,5,6]. Annual mean surface O_3_ concentrations have largely stabilized in Europe since the year 2000 as a result of emission control policies [7,8], but continued increase to 2050 is likely across South and East Asia [9,10]. The pattern of O_3_ exposure across regions is also expected to change over coming decades. Short peak ‘episodes’ of very high concentrations are predicted to become more frequent in India and China [10,11], while in Europe and North America a decline in peak episode frequency, alongside steadily increasing annual mean O_3_ concentrations, was observed between 1990 and 2010 [12]. Modelling suggests this decline in peak episode frequency in Europe and North America is likely to continue to 2050 [10].

Model-based estimates of yield loss under future climate and air pollution scenarios represent a powerful way of highlighting yield benefits that could come from reduced surface O_3_ [13,14,15,16]. Global O_3_-induced wheat yield loss for the year 2000 has been estimated as ranging from 5% to 26%, with potential additional losses of 1.5% to 10% predicted for 2030 [13,14]. However, nearly all large-scale assessments of O_3_-induced yield loss for wheat published to date have followed an empirical approach, where O_3_ concentration is simulated using a chemical transport model (CTM), concentration is linked to yield loss using response functions, and response is scaled up using crop production maps and agricultural statistics. An alternative is to use a process-based approach that could potentially produce more robust estimates of future yield, through inclusion of interactive effects between O_3_, CO_2_, and climate variables [17].

This process-based crop modelling approach builds on the development of methods for modelling O_3_ flux into leaves that have been made over recent decades [18,19]. These methods provide an hourly estimate of O_3_ dose reaching sites of damage in the leaf, creating the potential for O_3_ effects to be integrated into crop simulation models in a dynamic way. Studies applying O_3_ flux modelling have generally either used a multiplicative stomatal conductance (g_sto_) algorithm—described in prior research [20,21,22]—or followed a semi-mechanistic approach where g_sto_ is estimated empirically from photosynthetic rate, which in turn is modelled using the biochemical model of Farquhar et al. [23], also described in [24,25]. Since most crop models simulate growth responses at daily (and less frequently, hourly) timesteps and can respond to a changing environment [26], integration of O_3_ effects into crop models is feasible, if plant response to O_3_ can be represented in the model formulation. Attempts have been made to integrate O_3_ effects and crop modelling [25,27], but few estimates of O_3_-induced crop yield loss using a dynamic approach have been published to date [28,29]. Reasons for slow progress include the challenge of upscaling responses from the leaf to the canopy, the need for species and cultivar-specific model parameters, and the incomplete understanding of physiological mechanisms driving O_3_-induced yield reduction [17]. This paper goes some way to addressing these issues and identifying future research direction that would benefit from empirical investigation targeted towards developing the models that are currently in development.

It is well established that O_3_ exposure can reduce yield in wheat [30,31,32] and can cause foliar injury, impaired photosynthesis, altered carbon translocation, and accelerated senescence [33,34]. However, the processes linking O_3_ uptake to these responses are not fully understood and it is not clear which are most important in driving ultimate yield loss. Once O_3_ has been taken up through stomata, reactions in the plant apoplast lead to the formation of reactive oxygen species (ROS), which can then react with and damage membranes and proteins [35]. Most plants have in-built defence mechanisms and can up-regulate antioxidants to detoxify ROS, but this comes at a carbon cost, meaning O_3_ damage to productivity often occurs before visible symptoms appear [35,36]. An O_3_-induced reduction in photosynthetic rate has been widely reported [32,37,38], but quantifying the extent to which this is a direct effect of O_3_ on the photosynthetic mechanism, or indirect via changes to leaf pigmentation or g_sto_, has been a challenge for experimentalists. Disentangling direct O_3_ impacts on photosynthesis from the accelerated senescence response is also difficult. Some studies have observed reduced activity of the carbon-fixing enzyme ribulose-1,5-biphosphate carboxylase/oxygenase (rubisco) in response to O_3_ [39,40,41], leading to the hypothesis that ‘instantaneous’ effects of O_3_ on photosynthesis act via effects on this enzyme. The physiological mechanism underpinning the often-observed accelerated senescence response to O_3_ [42,43,44,45,46] is also unknown, although it has been hypothesized that it relates to long-term respiratory costs associated with detoxification and repair [25].

Several approaches for modelling O_3_ effects on photosynthesis and senescence have been published. In an early paper, Reich et al. proposed that ozone effects on a number of plant groups could be expressed via a linear relationship between exposure and growth [47]. Subsequent published approaches have attempted to model the separate effects of O_3_ on productivity and senescence and have tried to account for differential sensitivity across species. A function for modelling ‘instantaneous’ suppression of photosynthesis was proposed by Martin et al. [27], who simulated a linear reduction in carboxylation capacity of rubisco (the parameter V_cmax_ in the model of Farquhar et al. [23]), above a threshold hourly flux value representative of the species or cultivar-specific detoxification capacity. A similar approach of O_3_ acting on V_cmax_ was also used by Deckmyn et al. in their O_3_ damage module for forest trees, alongside an overnight repair mechanism and a parameter representing the respiratory cost of detoxification [48]. Ewert and Porter [25] applied a version of the Martin et al. ‘short-term’ function alongside a ‘long-term’ algorithm for modelling O_3_-induced senescence and assumed that ‘short-term’ photosynthetic suppression by O_3_ occurs throughout the leaf lifespan. Their senescence function assumes a linear reduction in mature leaf lifespan as accumulated O_3_ flux increases, and senescence comprises the final third of the mature leaf lifespan, during which time V_cmax_ is assumed to decline linearly. In this function, onset and completion of leaf senescence therefore move progressively earlier and closer together as accumulated O_3_ flux increases. An alternative approach for modelling O_3_-induced senescence is applied in the multiplicative DO_3_SE model (Deposition of ozone for stomatal exchange), a g_sto_ model which estimates accumulated O_3_ flux—known as the Phytotoxic Ozone Dose (POD)—to vegetation [18]. In this model, leaf senescence is induced by a threshold POD, which triggers curvilinear decline in leaf g_sto_ with a fixed shape but variable decline rate [20,49,50]. The POD ‘trigger’ can be parameterized according to the sensitivity of the cultivar or species.

The integration of O_3_ damage functions, such as those described above, into crop models could improve yield estimates under O_3_ stress. Model development must however be guided by experimental evidence that identifies damage mechanisms and their relative importance, informs parameterization, and indicates likely degree of error. Models must also be able to replicate the response to different patterns of O_3_ exposure. For example, acute peaks in concentration have been observed to induce greater yield loss than consistent, moderate levels with the same 24-h mean exposure [51], and modelling methods need to be able to capture these nuances.

This study combines data from two independent O_3_ exposure experiments on European wheat that took place at the same experimental facility in 2015 and 2016 and used consistent protocols for data collection. We model the accumulated O_3_ flux to wheat flag leaves across different O_3_ treatments, using the multiplicative DO_3_SE model, to derive the POD_0_SPEC metric of O_3_ flux exposure (mmol m^−2^ PLA day^−1^). Exposure to O_3_ flux is then considered alongside leaf chlorophyll, g_sto_, and photosynthesis responses in order to test key assumptions underpinning published O_3_ effect model functions. Firstly, with regards to O_3_ effects on leaf senescence, we (i) examine whether inter-cultivar differences in response are captured by current senescence functions and (ii) whether leaf senescence begins at an accumulated O_3_ flux ‘trigger’ value. Secondly, we examine whether O_3_ reduces V_cmax_ before—and therefore independent of—onset of O_3_-induced leaf senescence. Thirdly, we investigate if flux is a better predictor of the physiological response to O_3_ than concentration-based metrics and whether flux can account for differences in the pattern of O_3_ exposure (i.e., peak vs. background).

Our results reveal several insights about physiological responses to O_3_ that can add to the evidence base for designing O_3_ effect model functions. The two cultivars of European wheat studied here showed substantially different sensitivity to O_3_ in terms of their senescence response, indicating the importance of cultivar-specific parameterization in senescence functions. The study also finds that O_3_ effects on the photosynthetic mechanism are not observed in young flag leaves and are only seen following the onset of leaf senescence, suggesting that O_3_-induced accelerated senescence is more important than direct effects on photosynthesis in determining final yield.

## 2. Results

### 2.1. Ozone Treatments in 2015 and 2016

This study combines data from two independent experiments, which took place at the Centre for Ecology and Hydrology (CEH) air pollution exposure facility in 2015 and 2016. Both experiments gathered data on the physiological response of wheat cultivars to O_3_ exposure using consistent experimental design and data collection techniques. The main differences in these experiments were in the ozone concentration profiles to which plants were exposed. In 2015, two wheat cultivars, ‘Mulika’ and ‘Skyfall’, were exposed to eight O_3_ treatments. The treatments varied in their profile, with some characterized by acute peaks in concentration and others by a consistent background level (described in full in Section 4.1). In 2016, cv. ‘Skyfall’ was exposed to five O_3_ treatments, all with a ‘peak’ style profile. The treatments are named in this paper according to their severity and profile, these are ‘low background’ (LB), ‘low peak’ (LP), ‘medium background’ (MB), ‘medium peak’ (MP), ‘high background’ (HB), ‘high peak’ (HP), ‘very high background, (VHB), and ‘very high peak’ (VHP).

Table 1 summarizes the O_3_ treatments using several exposure indices including the 24-h mean (ppb) and seasonal AOT40 (ppm h). It also quantifies exposure using O_3_ flux, or phytotoxic ozone dose (POD), which was modelled in this study using the multiplicative DO_3_SE model (described in more detail in the Methodology and Appendix A). Several different flux metrics are included in the table. Because most plants have the ability to detoxify a small quantity of O_3_, methods for quantifying O_3_ dose often use a threshold for flux accumulation with the threshold value representing the detoxification capacity. Previous experimental work in wheat has found that a threshold of six produces the closest fit between exposure and yield [50]. We therefore applied a threshold of six when calculating accumulated flux, to produce the POD_6_SPEC metric (where SPEC refers to the species-specific nature of the metric). However, as thresholds for physiological effects are not as well established, we have also calculated accumulated flux with no threshold, known hereafter as the POD_0_SPEC. The mean daily POD_0_SPEC was also calculated, to represent the average level of exposure on a given day in each treatment.

### 2.2. Effect of O_3_ on Senescence 

Leaf senescence in the two cultivars across different O_3_ treatments was compared using chlorophyll content index (CCI) as a proxy measure for senescence. The growing seasons were divided into six segments or ‘bins’ of equal thermal time, and statistical analysis testing for the impact of O_3_ was carried out within each time-bin, in order to identify the point in the season when O_3_ effects on CCI can be seen to occur. CCI declined over the course of the growing seasons in both cultivars and in both years, and O_3_ accelerated this senescence (Figure 1). A substantial difference in senescence response of the two cultivars was observed. In the 2015 experiment, cv. Mulika exhibited O_3_-induced early senescence only in the highest treatment (VHP) (Figure 1A), whereas for cv. Skyfall in the same year, all treatments exhibited accelerated senescence relative to the lowest treatment (Figure 1B). In 2016, for Skyfall the three highest O_3_ treatments exhibited accelerated senescence (Figure 1C). A statistical summary of this analysis is reported in Table A3 of the Appendix A.

Analysis conducted within different thermal time groupings indicated that a significant negative effect of O_3_ on CCI was observed substantially earlier in the season for Skyfall compared to Mulika. For Skyfall in 2015, accumulated POD_0_SPEC was significantly negatively associated with flag leaf CCI from the third thermal time group onwards (1109–1337 °C days), after 25–36 days of O_3_ exposure. For Mulika in 2015, accumulated POD_0_SPEC was significantly negatively associated with CCI only at the fifth thermal time group (1568–1796 °C days), after 49–59 days of O_3_ exposure. The limited CCI data for Skyfall in 2016 supports the 2015 results, with a significant negative association between accumulated POD_0_SPEC and CCI observed in the 3rd and 4th thermal time-bins. A significant positive association between accumulated POD_0_SPEC and CCI was observed for Skyfall in 2015, in thermal time group one (649–879 °C days) spanning the first 14 days of O_3_ exposure. A statistical summary of the time-bin analysis is presented in Table A4 of the Appendix A.

The timing of leaf senescence onset and completion was influenced by O_3_ exposure. For both cultivars in 2015, leaf senescence onset occurred earlier in O_3_ treatments with higher mean daily POD_0_SPEC, although this trend was only statistically significant for Mulika (Figure 2A). On average, O_3_-induced senescence onset occurred later in the season for Mulika (1725 °C days) compared to Skyfall (1216 °C days). Senescence completion also occurred earlier in O_3_ treatments with a higher mean daily POD_0_SPEC in both cultivars, according to a linear relationship (Figure 2B). Senescence completion occurred at a similar thermal time on average for both cultivars (Mulika = 1841 °C days, Skyfall = 1867 °C days). The total duration of the O_3_-induced senescence period was therefore longer for Skyfall than Mulika. 

Skyfall also exhibited a linear reduction in the CCI-thermal time curve integral as the mean daily POD_0_SPEC increased (Figure 2C). This indicates that Skyfall exhibited reduced CCI in the flag leaf throughout the post-anthesis period in elevated O_3_. No significant association between mean exposure and curve integral was found for Mulika, although the highest treatment in terms of mean flux exposure (VHP) did exhibit a reduced integral compared to the other treatments.

### 2.3. Ozone Flux at Onset of Early Senescence

Leaf senescence data was also analysed to test whether a particular quantity of O_3_ flux, or ‘trigger’ flux value, could be identified as inducing the onset of leaf senescence. Accumulated O_3_ flux at leaf senescence onset for all 2015 treatments which exhibited significant accelerated senescence is shown in Table 2. In the highest O_3_ treatment (VHP), senescence onset occurred at a substantially lower accumulated POD_0_SPEC for the cultivar Skyfall (25.7 mmol m^−2^) compared with Mulika (30.1 mmol m^−2^), indicating differential sensitivity across the cultivars. When accumulated POD_0_SPEC and POD_6_SPEC at senescence onset are compared across the different O_3_ treatments for the cultivar Skyfall, senescence onset was observed to occur across a fairly wide range of accumulated flux (15.3–25.7 mmol m^−2^ POD_0_SPEC, 6.5–18.6 mmol m^−2^ POD_6_SPEC). The range of flux at senescence onset was more narrow when flux was calculated without a detoxification threshold (POD_0_SPEC flux range = 10.4 mmol m^−2^, POD_6_SPEC flux range = 12.1 mmol m^−2^) and considerably more narrow when only the five highest O_3_ treatments which exhibited the strongest accelerated senescence response are considered (POD_0_SPEC flux range of five highest treatments = 3.7 mmol m^−2^, POD_6_SPEC flux range of five highest treatments = 5.7 mmol m^−2^).

### 2.4. Response of Photosynthesis and g_sto_ Over Time and in Elevated O_3_

Figure 3 presents combined datasets for four leaf-level physiological parameters capable of short-term or ‘instantaneous’ change in response to environmental stimuli: *A_sat_* (Light-saturated photosynthetic rate), *V_cmax_* (maximum carboxylation capacity of rubisco), *J_max_* (maximum rate of electron transport) and *g_sto_* (stomatal conductance). Data from flag leaf measurements have been combined across both cultivars and both experiments, and the hue of each data point corresponds to the POD_0_SPEC that had accumulated at the time of measurement (an equivalent figure indicating accumulated POD_6_SPEC at time of measurement is shown as Figure A3 in the Appendix A). The average physiological values for high and low O_3_-treated plants within each time-bin are also shown on the plots. The average ‘low’ value represents the mean value for the lowest 2015 treatment (LB) and lowest 2016 treatment (LP2) combined. The average ‘high’ value represents the mean value for the highest 2015 treatment (VHP) and the highest 2016 treatment (VHP) combined. A decline in the photosynthetic parameters (*A_sat_*, *V_cmax_*, *J_max_*) was observed across the growing season, and this decline was accelerated in high O_3_. *g_sto_* did not decline over time in low O_3_ but did decline throughout the season in high O_3_.

The outcome of statistical analyses carried out on each cultivar–year combination and in each thermal time group, for the parameters *A_sat_*, *V_cmax_*, *J_max_,* and *g_sto_* as measured in the flag leaves, is shown in Figure 4. A statistical summary of this analysis is presented in Table A5, Table A6, Table A7 and Table A8 of the Appendix A. Grey regions on plots denote the period following the observation of a significant negative effect of accumulated POD_0_SPEC on flag leaf CCI. Across all cultivar–year combinations, no significant negative effects of accumulated POD_0_SPEC on any of the instantaneous physiological parameters was observed before negative effects of accumulated POD_0_SPEC on CCI were observed. A significant negative association of accumulated POD_0_SPEC on the parameters *V_cmax_* and *J_max_* was not observed until the 5th thermal time-bin (1568–1796 °C days).

Some evidence of heightened physiological performance in the early-season in high O_3_ was observed across both cultivars and both years, although the pattern was not consistent. A significant positive association between accumulated POD_0_SPEC and physiology in either the first or second time-bin was observed for i) *J_max_* in Mulika in 2015, ii) *A_sat_* and *g_sto_* in Skyfall in 2015, and iii) *A_sat_* and *V_cmax_* in Skyfall in 2016.

### 2.5. Comparison of O_3_ Exposure Metrics for Predicting Physiological Response to O_3_

For all measured physiological parameters (CCI, *A_sat_*, *V_cmax_*, *J_max_*, *g_sto_*) and for both cultivars, a flux-based metric of exposure was better at predicting physiological response of wheat to O_3_, compared to the concentration-based metrics (24-h mean and AOT40) (Table 3). For four out of the ten model sets created in this analysis, the accumulated POD_0_SPEC (i.e., without a threshold for accumulation) produced the best model fit. For the other six model sets, the accumulated POD_0_SPEC and POD_6_SPEC metrics were equally good at predicting physiological response. The O_3_ flux metric with no threshold for accumulation was therefore equal to, or better than, the O_3_ flux metric with a detoxification threshold at predicting the physiological response to O_3_.

The inclusion of an explanatory variable describing the profile (i.e., peak or background) of O_3_ exposure in the ‘best’ model did not improve fit in nine out of the ten model sets created with the accumulated POD_0_SPEC metric, and in all models created with the accumulated POD_6_SPEC metric. Using O_3_ flux as the metric of exposure therefore accounts for differences in the O_3_ exposure resulting from peak-dominated treatments and those featuring a consistent background level, in the majority of cases.

## 3. Discussion

The first aim of the analysis presented here was to assess whether published approaches for modelling O_3_-induced senescence can account for inter-cultivar variation in response. Both cultivars exhibited accelerated senescence in response to O_3_, but the pattern of response differed according to cultivar. In 2015, significant accelerated senescence was observed in seven O_3_ treatments for Skyfall, but only in the highest treatment for Mulika, suggesting higher O_3_ tolerance in Mulika (Figure 1). This differential tolerance is also indicated by the earlier appearance of significant O_3_ effects on leaf CCI across all treatments for Skyfall compared to Mulika. Senescence completion occurred progressively earlier, hence total leaf duration became progressively shorter, in both cultivars as average O_3_ flux (mean daily POD_0_SPEC) in the treatment increased (Figure 2B), according to a linear relationship. Completion of leaf senescence occurred at a similar thermal time in both cultivars (Mulika = 1841 °C days, Skyfall = 1867 °C days), meaning that the total senescence duration was longer for Skyfall. While O_3_-induced senescence in Mulika was characterised by a sudden drop in leaf CCI in the late-season, Skyfall exhibited a more gradual O_3_-induced decline in CCI.

The linear relationship between mean flux and total leaf duration observed in this study for both cultivars gives support to the senescence function of Ewert and Porter [25], which assumes a linear decline in mature leaf lifespan as O_3_ exposure increases. However, evidence presented here suggests that the duration of leaf senescence is likely to vary with O_3_ exposure. Additionally, the differential senescence duration in the two cultivars suggests that a key assumption of the Ewert and Porter function—that leaf senescence comprises the final third of the mature leaf lifespan—may not hold true for all cultivars. For example, in 2015 for Skyfall, leaf senescence in the highest O_3_ treatment comprised 76.7% of the total flag leaf lifespan (flag leaf emergence = 877 °C days, leaf senescence onset = 1075 °C days, senescence completion = 1725 °C days)—substantially greater than one third. The inter-cultivar variation in senescence response observed in this study would therefore only be captured by a model function that allows for the proportion of leaf lifespan comprising leaf senescence to be parameterised according to cultivar and ozone exposure. Our results suggest that to effectively model variation in the pattern of O_3_-induced senescence, the timing of senescence onset, and the rate (or duration) of senescence, models need to be calibrated for particular species and cultivars.

However, the analysis in this paper highlights the uncertainty associated with the approach of Danielsson et al. [20] for modelling onset of O_3_-induced senescence using a threshold of accumulated flux. Following this approach, senescence onset may occur at different points in time at different levels of mean exposure but should occur at approximately the same value of accumulated flux. This method was designed in the absence of a known mechanism for induction of senescence by O_3_ but could be interpreted mechanistically if accumulated O_3_ flux is assumed to be proportional to increased respiratory effort accumulated over the season, which has been proposed as a potential trigger for O_3_-induced senescence [25]. For Skyfall, across the five highest O_3_ treatments in 2015, onset occurred across a POD_0_SPEC range of 22.0–25.7 mmol m^−2^. Given the limitations associated with the method used to identify senescence onset—arbitrarily defined as a 10% reduction in leaf CCI relative to the control—as well as the inherent variation that exists between seedlings, this flux range can be considered relatively narrow. However, when all treatments that exhibited a significant O_3_ effect on senescence are considered for Skyfall in 2015, the range of flux at senescence onset is considerably wider (17.8–25.7 mmol m^−2^ POD_0_SPEC). These results provide an estimate of the degree of error potentially associated with applying this approach in models and suggest that the accumulation of respiratory effort does not fully explain the triggering of O_3_-induced senescence.

A second objective of this study was to test the hypothesis that O_3_ reduces photosynthetic rate in the short term by reducing carboxylation capacity of rubisco (*V_cmax_*). The assumption that O_3_ reduces *V_cmax_* is central to the ‘instantaneous’ O_3_ effect function of Martin et al. [27]. A version of this function is also applied by Ewert and Porter [25], where O_3_ reduces photosynthesis in a short-term and reversible way, in addition to and independent of the O_3_ senescence effect. The analysis presented here found that a significant negative effect of O_3_ on photosynthesis and *g_sto_* was only observed concurrent with O_3_-induced leaf senescence (Figure 4). This result was consistent across all cultivar–year combinations. For Skyfall in 2015 and 2016, significant negative effects of O_3_ on *A_sat_* were observed *before* a negative association between O_3_ and *V_cmax_*, suggesting that reduced carboxylation capacity is not responsible for the initial reduction in photosynthetic capacity observed in these experiments. There was therefore no evidence of an ‘instantaneous’ effect of O_3_ on the photosynthetic mechanism, in the period preceding leaf senescence.

These results contradict several studies which observed short-term reduction in photosynthetic rate in response to O_3_ [39,40,41]. One possible explanation for this contradiction is that instantaneous reduction of carboxylation capacity by O_3_ is only relevant at acute concentrations. The reduced carboxylation efficiency reported by Farage et al. [41] was observed following 4–16 h of exposure at unrealistically high O_3_ concentrations of 200–400 ppb, considerably higher hourly concentrations compared to those used in the experiments in this study, which more closely mimic ambient conditions (maximum hourly O_3_ exposure of 117 ppb). The results presented here therefore indicate that accelerated senescence is likely to be more important than direct effects on photosynthesis for determining crop yield loss in most agricultural landscapes, where O_3_ concentrations are typically moderate (ranging from ~20 to ~45 ppb) with occasional peaks in concentration [5,52]. Understanding and simulating the early senescence response to O_3_ should therefore be the priority for O_3_ experimentalists and modellers.

Alternatively, our results could be explained by a differential response to O_3_ in younger and older leaves. Bernacchi et al. [53] and Morgan et al. [54] observed in field experiments with soybean that O_3_ effects on photosynthesis and *g_sto_* were not apparent in new fully expanded leaves and Reichenauer et al. [55] saw similar results in three wheat cultivars. Younger leaves may have a higher tolerance to O_3_, or alternatively the O_3_ effect on photosynthesis may be associated with a cumulative build-up of O_3_ damage in leaves or leaf age. Either way, the age-dependency of O_3_ effects is an important consideration in O_3_ effects modelling. The function described by Ewert and Porter [25] for modelling short-term effects of O_3_ on photosynthesis allows for leaf age to influence the rate of overnight recovery from O_3_ damage but not the threshold for damage. The role of leaf age in determining O_3_ flux thresholds would benefit from further investigation.

A surprising result from the data analysis is that O_3_ had a significant positive effect on some physiological parameters early in the season. CCI, *A_sat_*, *V_cmax_*_,_ and *g_sto_* all exhibited a positive association with O_3_ exposure for one or more of the cultivar–year combinations, in either the first or second thermal time group (up to 32 days following beginning of exposure). Stimulation of growth at low doses of a toxin is known as hormesis, and this phenomenon has been observed previously in a number of plants in response to exposure to low concentrations of O_3_ [56]. Stimulation of photosynthesis and *g_sto_* in wheat during the first few weeks of O_3_ exposure was also observed by Mulholland et al. [57] in their open-top chamber experiment, although generally there are few reported cases of this phenomenon in crop species. Ozone-induced physiological stimulation could be an adaptive response associated with plant defence responses, for example, heightened *g_sto_* and photosynthesis may enable the upregulation of antioxidant synthesis. Observations that yield can be stimulated at low O_3_ exposure concentrations have also given rise to the theory that free radicals, at low concentrations, can act as growth promotors in plants [58]. An alternative hypothesis is that the observed early-season physiological boost in this study is related to disruption of stomatal control by O_3_, as has been observed in some grassland species [59,60], leading to heightened *g_sto_* and an associated boost in other physiological parameters. Our results suggest that a bi-phase hermetic dose-response curve, such as proposed by Agathokleous et al. [56], could be considered in ozone effect functions. More experimental data is needed in order to establish if the early-season physiological boost induced by O_3_ in this study is consistent across other plant species and environments.

The third aim of this study was to test whether O_3_ flux would be a better predictor of physiological response than concentration-based exposure (AOT40 and 24-h mean). Flux was superior at predicting the response to O_3_ of five physiological variables (CCI, *A_sat_*, *V_cmax_*, *J_max_*, *g_sto_*) in regression models, for both cultivars (Table 3). Previous studies have reported that flux is better than AOT40 at predicting the spatial distribution of O_3_ injury [61] and is a better predictor of the response of wheat yield, poplar biomass, and assimilation rate in urban trees to O_3_ exposure [62,63,64]. However, few have compared the association between leaf-level physiology and different exposure metrics in crop species. Our results align with the general consensus in the O_3_ research community that O_3_ flux represents a more biologically relevant metric of O_3_ exposure than ambient concentration [61,65,66,67] and indicate that O_3_ flux should be the preferred metric of exposure in O_3_ effect model functions. More surprising is the fact that the flux metric without an accumulation threshold, POD_0_SPEC, was a better or equal predictor of physiological response compared to POD_6_SPEC, which employs an accumulation threshold of six. POD_6_SPEC produced the closest correlation between flux and yield of wheat in a previous analysis testing different flux accumulation thresholds [50] and has been applied in several assessments of O_3_ impacts in wheat [22,68]. More research is therefore needed to establish how much the capacity to detoxify O_3_ varies between cultivars and why the threshold flux required to induce leaf-level physiological changes appears to differ from the threshold required to reduce yield. This result also highlights the limitations of an empirical rather than a mechanistic approach to modelling ozone tolerance and detoxification capacity. The development and integration of mechanistic ozone detoxification modules in models, such as that proposed by Plöchl et al., could potentially lead to improved model accuracy [69].

The view that O_3_ flux should be the metric of exposure in O_3_ effect modelling is also supported by the fact that O_3_ flux accounted for the different levels of exposure in treatments dominated by peaks in concentration, versus those characterised by a consistent background level, for the majority of physiological parameters. Flux is therefore likely to perform well as a predictor of physiological response across different world regions which are currently experiencing divergent trends in the pattern of O_3_ exposure [10,11,12].

Limitations of this study need to be considered when interpreting and applying the results. Calculated values of O_3_ flux were not verified by leaf-level measurements of gas flux through stomata. However, the decision to apply the multiplicative DO_3_SE model in this study was based on the fact that fluxes produced by this model have previously been evaluated in several independent studies which have demonstrated the model’s predictive capability [19,21,22,70,71,72]. A further limitation is that estimates of the onset of leaf senescence were based on leaf chlorophyll content, which would have represented both the chlorophyll loss resulting from leaf injury, as well as chlorophyll loss relating to senescence. In addition, the analysis is based on only one crop species and two cultivars. As the variation in yield response to O_3_ exhibited by different crops, and different cultivars within the same crop species, is well established [2,73], caution must be used when extrapolating results presented here to other wheat cultivars and crops. It should however be noted that the observation in this study that no O_3_ effect on photosynthesis could be observed in young wheat leaves—indicating the senescence response is more important than direct effects on photosynthesis—is supported by previous work in other wheat cultivars [55] and by other experimental work in soybean [53,54]. Considerations when applying the results presented in this study, particularly when attempting to up-scale modelled responses from the leaf to canopy level, include the fact that the response observed in the wheat flag leaf may differ from the responses of lower-canopy leaves, and exposure to O_3_ during early seedling and leaf development may also alter the sensitivity to O_3_ observed in the flag leaf.

In conclusion, this study has shown that current approaches for modelling O_3_ effects on leaf longevity and photosynthesis in crops have some limitations and are not fully supported by the experimental data presented here. Model functions representing O_3_-induced senescence must allow for parameterisation of the timing of senescence onset and rate of senescence, if inter-cultivar variation in response is to be accurately simulated. Further research aimed at understanding the mechanistic ‘trigger’ of O_3_-induced senescence should be a priority, as this understanding may allow for the development of a more effective mechanism in models for inducing the senescence response. The results also suggest an age-dependency in the response of photosynthesis to O_3_—which is not currently fully considered in modelling methods—and indicate that acceleration of senescence is more important than direct effects of O_3_ on photosynthesis in determining final O_3_-induced yield loss, at O_3_ concentrations that crops are likely to be exposed to on a day-to-day basis. Building functions that can accurately represent the O_3_-induced senescence effect in crops should therefore be the priority for O_3_ effect modellers.

## 4. Materials and Methods

### 4.1. Experimental Site and Treatments

Both experiments that provided data for this study took place at the CEH air pollution exposure facility in Abergwyngregyn, North Wales (53.2°N, 4.0°W). The two experiments were conducted independently and differ in terms of the number of cultivars and treatments, however, the experimental design was similar across both years and the same protocols for physiological measurements were applied. Data was therefore subsequently pooled across the two years to produce a more robust dataset. In 2015, two European wheat cultivars (*Triticum aestivum* L., ‘Mulika’ and ‘Skyfall’) were exposed to O_3_ for 82 days. In 2016, only the more O_3_-sensitive cultivar, ‘Skyfall’, was used and was exposed for 92 days. Timelines for sowing, emergence, O_3_ exposure, and harvest in both experiments are presented in Figure A1 of the Appendix A. In both experiments, plants were grown in 25-litre containers (40 cm × 35 cm × 38 cm) filled with John Innes No.3 compost (Westland Horticulture Limited, Huntingdon, UK), and soil was inoculated shortly after sowing with microbial communities using a soil slurry taken from a nearby wheat field. Seeds were sown in rows 7 cm apart at a density of ~260 seedlings per square metre, which aligns with recommended seedling density for field conditions [74]. Four containers per cultivar–treatment combination were planted and placed alongside each other, producing a canopy of ~144 plants per cultivar–treatment combination. In both years, ammonium nitrate fertiliser was applied once mid-season (80 kg/ha). In 2015, fungicide (‘Unix’, cyprodinil, 1.6 kg/ha—Syngenta, Bracknell, UK) was applied once and insecticide (pyrethrum, 1 mL/L—Bayer, Monheim Am Rhein, Germany) applied three times. In 2016, fungicide was applied twice (1st application: Trifloxystrobin, 0.12 g/L—Bayer, Mondheim Am Rhein, Gernany; tebuconazole, 0.125 g/L—Bayer, Mondheim Am Rhein, Germany; 2nd application: cyprodinil, 2.25 g/L—Syngenta, Bracknell, UK) and insecticide applied once (thiachloprid, 0.15 g/L #x2014;Bayer, Mondheim Am Rhein, Germany).

Ozone exposure took place within ‘solar domes’, hemispherical glass domes three metres in diameter and two metres in height, described previously [59,75]. Air entering domes was carbon-filtered to remove O_3_ and other air pollutants (i.e., NO_X_ and SO_2_), and the solar dome site is an area with some of the lowest NO_X_ and SO_2_ emissions in the country and is not downwind of any major sources [76], minimising any other pollutants that could have entered the domes after filtration. Following filtration, a precision-controlled quantity supplied by an O_3_ generator (Dryden Aqua G11, Edinburgh, UK) linked to an oxygen concentrator (Sequal 10, Pure O2, Manchester, UK), was added to incoming air. Injection concentrations were determined by a computer-controlled O_3_ injection system (Lab VIEW, version 8.6, National Instruments, Austin, TX, USA). Air within domes was circulated at a rate of two air changes per minute, and the O_3_ concentration within each dome was recorded on a 30-min cycle using two O_3_ analysers of matched calibration (Envirotech API 400A, St Albans, UK). Exposure profiles for each treatment are presented in Figure 5. Treatments spanned a range of seasonal mean concentrations and represented different O_3_ exposure patterns, representing potential future profiles of increasing background or decreasing peak O_3_. In 2015, four treatments consisted of a low night-time background level, with high peaks during the day, classified as ‘peak’ treatments, while the other four treatments comprised of consistent concentrations with only small peaks, classified as ‘background’ treatments. In 2016, all treatments were ‘peak’ in profile. Although O_3_ treatments were not replicated, numerous studies have established the statistical validity of conducting unreplicated experiments using the solar dome facility [56,77,78], and previous work has shown that no solar dome effect on air or leaf temperature is detectable [79].

Climatic conditions fluctuated naturally in the solar domes according to ambient conditions. Air temperature, photosynthetically active radiation (PAR), relative humidity, and wind speed were monitored in one solar dome during both experiments using an automatic weather station (Skye instruments Ltd, Llandridod Wells, UK) to obtain data for stomatal flux modelling. Plants were well-watered throughout, and soil moisture content was continuously monitored in selected plant containers to a depth of 10 cm using Theta Probes (Delta-T Devices Ltd., Cambridge, UK).

### 4.2. Leaf Chlorophyll and Gas Exchange Measurements

Chlorophyll content was measured non-destructively as an index (chlorophyll content index, CCI) using CCM-200 and CCM-200+ instruments (Opti-sciences, Hudson, NH, USA). A regression line fit to paired measurements was used to standardise observations made using the two instruments. In 2015, 684 measurements were made over 70 days; in 2016, 105 measurements were made over 22 days.

To assess the effect of O_3_ on photosynthetic capacity and *g_sto_*, response curves of net photosynthetic rate (*A*) to intercellular CO_2_ concentration (C_i_), i.e., *A*–C_i_ curves, were constructed using a portable infrared gas analyser (Li-Cor 6400XT; LI-COR Biosciences, Lincoln, NE, USA). In 2015, measurements were made in the two lowest O_3_ treatments at the beginning of exposure (20–26 May). Further measurements in the two lowest treatments (LB and LP) and two high O_3_ treatments (VHB and VHP) were made in the mid-season (8–17 June) and late-season (16–24 July). Measurements were made in the youngest fully expanded leaf of randomly selected plants (represented by the flag leaf) from 28th May onwards. In 2016, four sets of *A*–C_i_ curve measurements were made at approximate two-weekly intervals spanning 6 June to 29 July. Measurements in 2016 were made in all treatments at each of the time intervals, except for the final measurement set in late July, when plants in HP and VHP treatments were too senesced for measurements to take place. All 2016 measurements were made in the flag leaf. For both years, four *A*–C_i_ measurements were made per treatment and per cultivar at each timepoint, and leaves were tagged following measurement so that the same leaf could be measured throughout the season.

All response curve measurements were conducted at a selected light saturation (minimum photosynthetic photon flux density = 1500 µmol m^−2^ s^−1^, LED light source), and sample chamber relative humidity was maintained between 50 and 80%. Photosynthetic rate and *g_sto_* were allowed to stabilise in the leaf chamber at ambient CO_2_ (400 µmol mol^−1^). The *A*–C_i_ curve was constructed by measuring *A* at a minimum of nine air CO_2_ concentrations, ranging from ca. 50 to 2000 µmol mol^−1^. *A*_sat_ and associated *g_sto_* values were determined from the ambient CO_2_ measurements (400 µmol mol^−1^) from each *A*–C_i_ curve.

Additional measurements of *A*_sat_ and associated *g_sto_* were made in 2016 over six days (16 June, 1 July, 8 July, 14 July, 20 July, 26 July). Measurements were made at ambient CO_2_ concentration (400 µmol mol^−1^) under the same light and relative humidity conditions as described above.

### 4.3. Derivation of V_cmax_
*and* J_max_

Maximum rate of carboxylation (*V_cmax_*) and maximum rate of electron transport (*J_max_*) were derived from *A*–C_i_ curves using the estimating utility and methodology described by Sharkey et al. [80]. Leaf temperature and atmospheric pressure were the input parameters, which were measured using the Licor 6400XT simultaneously with all photosynthesis measurements. *V_cmax_* and *J_max_* values calculated from curves were adjusted to 25 °C.

The *V_cmax_* dataset was extended by applying the ‘one-point method’ of deriving *V_cmax_* from *A_sat_* as described by De Kauwe et al. [81]. Estimation of *V_cmax_* when only *A_sat_* is known using the one-point method relies on the assumption that photosynthetic rate at ambient CO_2_ is rubisco-limited [81]. As the measurements of *A* at 400 µmol mol^−1^ CO_2_ in the measured *A*–C_i_ curves typically fell within the rubisco-limited section of the curve (i.e., before the transition point), this assumption was thought to be likely to hold true for the two cultivars used in this study. The one-point method also assumes, in the absence of a known daytime respiration rate (*R_day_*), that *R_day_* can be estimated as 1.5% of *V_cmax_*. *V_cmax_* was calculated from *A_sat_* using the following equation:(1)$$V_{cmax}=A_{sat}*\left(\frac{C_{i}+K_{m}}{C_{i}-\;\Gamma *}-0.015\right)$$
where *K_m_* is the Michaelis–Menten constant, given by:(2)$$K_{m}=K_{c}\;*\;\left(1+\frac{O_{i}}{K_{O}}\right)$$

The parameters *K_c_* (Michaelis–Menten constant for CO_2_), *K_O_* (Michaelis–Menten constant for O_2_) and Γ* (CO_2_ compensation point in the absence of mitochondrial respiration) were estimated at 25 °C following the equations and constants published by Bernacchi et al. [82] describing their temperature dependence in the model species tobacco (*Nicotiana tabacum,* L.). Equations and constants used to derive these three parameters are listed in Table A2 of the Appendix A. O*_i_* represents the intercellular concentration of O_2_ (210 mmol mol^−1^) [81].

The robustness of the one-point method was evaluated by comparing *V_cmax_* values calculated from a subset of the measured *A*–C_i_ curves with the *V_cmax_* values calculated from each corresponding *A_sat_* value (i.e., the 400 µmol mol^−1^ CO_2_ value from each *A*–C_i_ curve). *V_cmax_* values derived using both methods were adjusted to 25 °C. A very close association was observed between *V_cmax_* values derived using the two methods (Figure 6, adjusted *r^2^* = 0.95, *p* < 0.001), indicating that the one-point method is robust for the cultivars used in this study. *V_cmax_* values derived using the one-point method were therefore pooled with *A*–C_i_-derived *V_cmax_* values for analysis, and the potential error introduced through the use of two different derivation methods was accounted for in the statistical analysis by including—in model selection—an explanatory variable describing the derivation method (explained in more detail in Section 4.7.3).

### 4.4. Modelling O_3_ Flux

Stomatal O_3_ flux to the flag leaf was modelled in each treatment for both years to derive a measure of exposure that accounted for the environmental influence on O_3_ uptake and could be tracked over time. Flux was modelled using the multiplicative *g_sto_* module of the DO_3_SE model [18], which has a published parameterisation for European wheat [49,50,83] and has been applied previously to model O_3_ flux to this crop [22,84]. A summary of the DO_3_SE algorithms and parameters used in this study are presented in Appendix A.

Ozone flux for wheat is accumulated above a detoxification threshold of six in the DO_3_SE methodology (producing the POD_6_SPEC flux metric, species-specific phytotoxic O_3_ dose above a threshold of 6 mmol m^−2^ PLA s^−1^—previously known as the POD_6_, with “SPEC” referring to the species-specific version of the DO_3_SE model) [83], as this threshold has produced the closest correlation between POD and wheat yield in previous experiments [50]. However, as thresholds of physiological effect in wheat have been far less studied, the POD_0_SPEC (where no threshold for accumulation is applied, previously known as the POD_0_) was also calculated, in order to avoid assuming a threshold of effect. Modelled POD_0_SPEC and POD_6_SPEC for 2015 and 2016 O_3_ treatments are shown in Figure 7.

### 4.5. Alignment of Physiological Observations with O_3_ Flux, and Calculation of Mean Flux Exposure (Mean Daily POD_0_SPEC)

Each physiological observation (CCI, *A_sat_, V_cmax_, J_max_, g_sto_*) was aligned with the treatment-specific accumulated POD_0_SPEC and POD_6_SPEC on the day of measurement and at the exact time of measurement, wherever this data was available (referred to hereafter in this paper as ‘accumulated POD_0_SPEC’ and ‘accumulated POD_6_SPEC’). This was done to allow the impact of real-time O_3_ flux exposure on physiology to be analysed. The mean daily POD_0_SPEC (i.e., the average accumulation of flux per day, mmol m^−2^ PLA day^−1,^) was also calculated for each O_3_ treatment to act as a metric of mean exposure intensity. Mean daily POD_0_SPEC values for each O_3_ treatment are presented in the results section in Table 1. Mean daily POD_0_SPEC was calculated as the average of daily POD_0_SPEC accumulation from *A_start_* until the modelled onset of senescence.

### 4.6. Data Standardisation

The two experiments had different sowing and harvest calendars. In order to compare timings across the two experiments, time was therefore standardised by calculating thermal time from plant emergence onwards (daily mean temperature sum >0 °C). Physiological data was also standardised, by conversion from raw to relative values. This was done to account for differences in instrument calibration between years, and to account for differences in beginning of season ‘baseline’ physiology between cultivars. Relative values for the physiological observations were calculated by deriving a reference value for each parameter (CCI, *A_sat_, V_cmax_, J_max_, g_sto_*) and each cultivar–year combination (i.e., Mulika in 2015, Skyfall in 2015, Skyfall in 2016). The reference value—calculated as the 90th percentile value of all observations, spanning the whole season and all treatments—was used as the baseline for calculating relative change. Skyfall CCI data from 2016 comprised too few data points for the derivation of an individual reference value, 2016 and 2015 CCI data for Skyfall was therefore combined to produce a single reference value for Skyfall, as CCI data for Skyfall was found to not significantly differ by year (*p* = 0.06 in regression model). A comparison of CCI data for Skyfall measured in 2015 and 2016 can be found in Appendix A.

### 4.7. Statistical Analysis

Statistical analysis was conducted in R version 3.3.2 [85], and either involved linear regression or linear mixed models (LMMs) using the package lme4 v1.17. Model selection was by AIC (Akaike Information Criterion). The model with the lowest AIC was considered the ‘best’ model of those fitted, and models differing in <2 AIC units from the best model were defined as having little empirical support [86]. Wherever relevant, a random factor describing solar dome number was included in models to account for multiple measurements made within domes, and unique pot ID was a random factor when analysis involved multiple measurements made from the same pot. *p*-values were obtained for terms in the optimal models using the R package lmerTest, v2.0-33 [87]. Assumptions of normality and even spread of residuals were checked using residual plots and data were transformed where necessary. Four key analyses were conducted as part of this study and are described in more detail below.

#### 4.7.1. Identification of O_3_ Treatments with Significantly Accelerated Senescence

Flag leaf CCI data was analysed in all O_3_ treatments to test for accelerated senescence. Each elevated O_3_ treatment was paired in turn with the control treatment for that experiment, and the significance of the thermal time/mean daily POD_0_SPEC interaction term was tested using LMMs. Control treatments were defined as the lowest in terms of mean daily POD_0_SPEC and comprised of treatment LB for the 2015 experiment and treatment LP2 for the 2016 experiment.

#### 4.7.2. Analysis of O_3_ Effect on the Timing of Senescence Onset and Completion

The impact of O_3_ on leaf senescence onset and completion was examined using regression models fitted to each of the 2015 O_3_ treatments (separately for the two cultivars). It was not possible to conduct this analysis for 2016, as 2016 CCI measurements only spanned 22 days. Regression models comprised of relative CCI as the dependent variable and thermal time as the independent variable, and the shape of response was determined by comparing linear, quadratic, and cubic models. The best model for each O_3_ treatment was then used to determine i) thermal time at leaf senescence onset, ii) thermal time at senescence completion, and iii) the post-anthesis curve integral (i.e., area under the curve), as shown in Figure 8. Thermal time at senescence onset was aligned with the accumulated POD_0_SPEC at that time for each O_3_ treatment, to identify the accumulated flux ‘trigger’ values for senescence onset.

#### 4.7.3. Analysis of Relative Timing of O_3_ Effects on Different Aspects of Physiology

The effect of accumulated POD_0_SPEC on CCI, *A_sat_*, *V_cmax_*, *J_max_*, and *g_sto_* during successive periods of the growing season was analysed, to identify when O_3_ began to influence physiology. The range of thermal time spanned by flag leaf physiological measurements was divided into six thermal time-bins of equal width. The effect of accumulated POD_0_SPEC on each parameter, within each time-bin and for each cultivar–year combination, was analysed by comparing model fit with and without accumulated POD_0_SPEC as an explanatory variable. An additional explanatory variable was included in model selection for *V_cmax_*, describing the derivation method (i.e., A–Ci curve or one-point method).

#### 4.7.4. Comparison of Flux and Concentration-Based O_3_ Exposure Metrics for Predicting Physiological Response

Accumulated POD_0_SPEC, accumulated POD_6_SPEC, 24-h mean concentration, and AOT40 (accumulated O_3_ > 40 ppb during daylight hours) were compared in their ability to predict the response of CCI, *A_sat_*, *V_cmax_*, *J_max_*, and *g_sto_* during the 5th thermal time-bin. The 5th time-bin was selected for this analysis as most physiological parameters exhibited a response to O^3^ exposure during this time. For each physiological parameter, LMMs constructed with each of the metrics of O_3_ exposure were compared for model fit. An explanatory variable describing whether O_3_ had been administered as a ‘peak’ or ‘background’ profile was also included in model selection, to test whether the O_3_ metric that produced the best model fit also accounted for different patterns of exposure.

## Figures and Tables

**Figure 1 plants-08-00084-f001:**
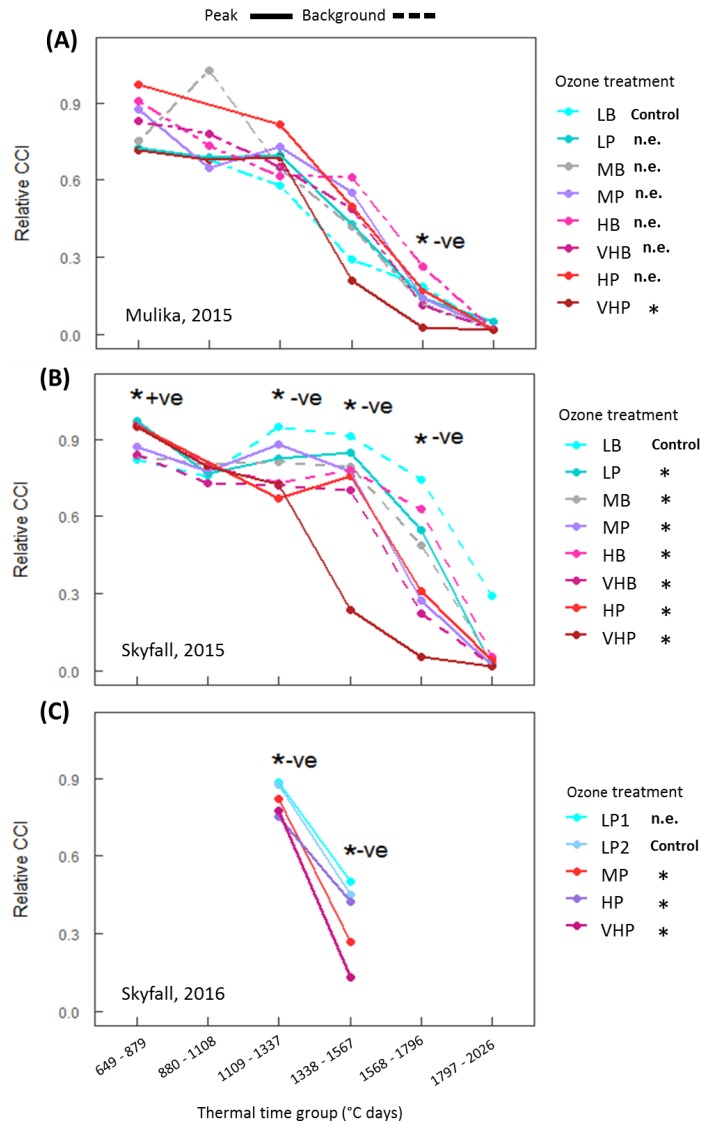
Average relative chlorophyll content index (CCI) in flag leaves for six thermal time groups, (**A**) cv. Mulika in 2015, (**B**) cv. Skyfall in 2015 and (**C**) cv. Skyfall in 2016. Time-bins where a statistically significant association between CCI and accumulated POD_0_SPEC was observed are marked with an asterisk (*). The direction of O_3_ effect, i.e., positive (+ve) or negative (−ve) effect on CCI, is also shown. Ozone treatments which exhibited a significant early decline (defined as a decline in CCI of 10% or more relative to the control treatments) are marked in the figure keys with asterisks (*), and those which showed no effect are marked as n.e. (no effect).

**Figure 2 plants-08-00084-f002:**
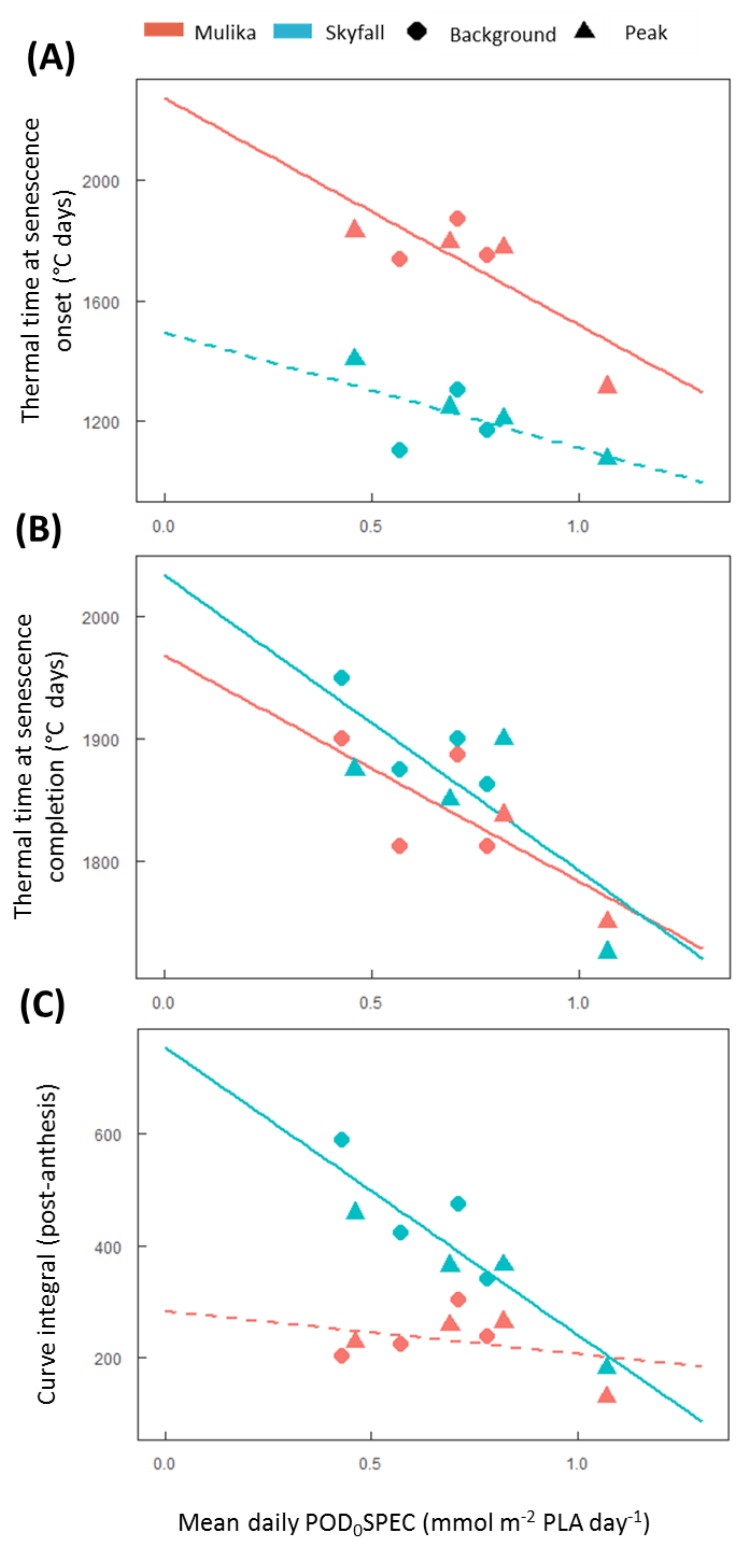
Effect of O_3_ on the onset and completion of leaf senescence in 2015. (**A**) Thermal time at senescence onset versus the mean daily POD_0_SPEC in each treatment. (**B**) Thermal time at senescence completion versus mean daily POD_0_SPEC in each treatment. (**C**) Area under the post-anthesis section of the CCI-thermal time curve versus the mean daily POD_0_SPEC in each treatment. Solid trend lines indicate a significant regression (*p* < 0.05), dashed lines indicate that the trend was not significant.

**Figure 3 plants-08-00084-f003:**
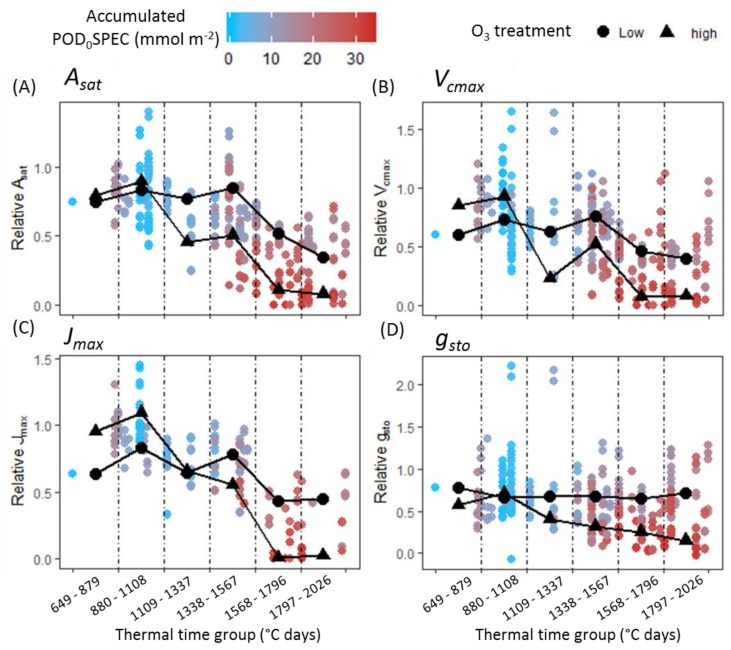
Flag leaf data for (**A**) *A_sat_*, (**B**) *V_cmax_*, (**C**) *J_max_*, and (**D**) *g_sto_*, combined across all cultivar–year combinations. The hue of each data point corresponds to the accumulated POD_0_SPEC at the moment of measurement. Mean values of physiological parameters in low O_3_-treated plants (averaged across 2015 LB and 2015 LP2 treatments) and high O_3_-treated plants (averaged across 2015 VHP and 2016 VHP treatments) are shown as black data points on the plots.

**Figure 4 plants-08-00084-f004:**
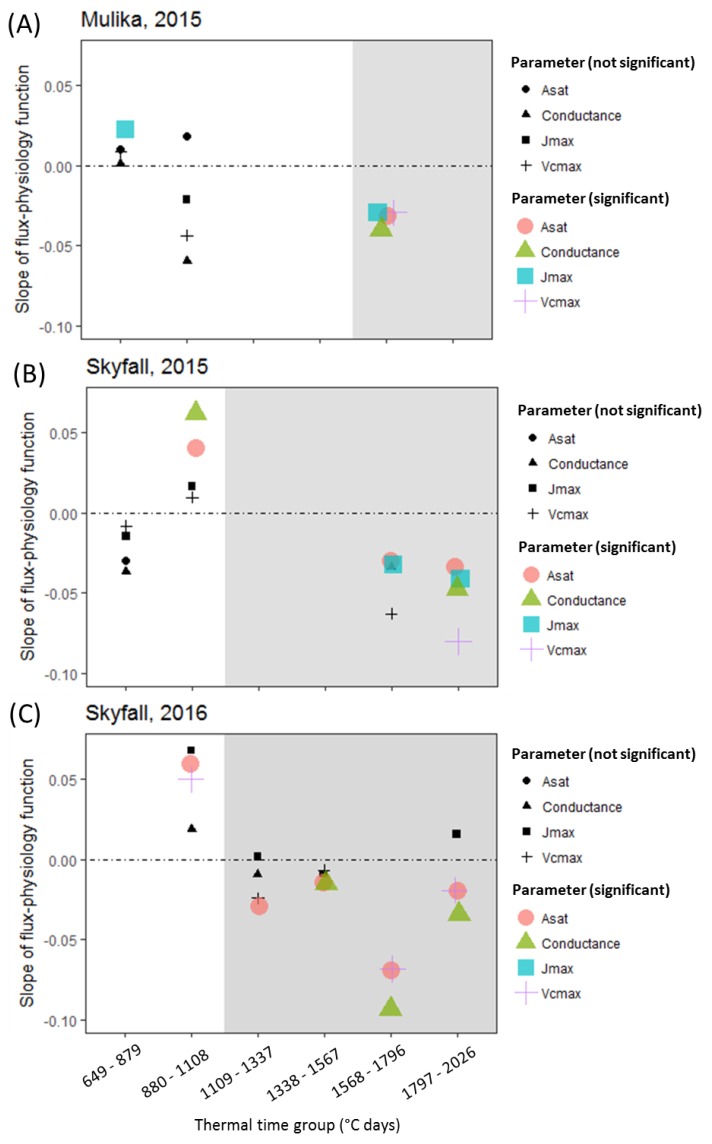
Plots showing the response of *A_sat_*, *V_cmax_*, *J_max_*, and *g_sto_* to O_3_ flux. The y-axis represents the accumulated POD_0_SPEC-physiology slope in the ‘best’ linear mixed regression model for each thermal time group. Positive slope indicates a positive effect of O_3_ on the physiological variable; a negative slope indicates a negative effect. (**A**) cv. Mulika in 2015, (**B**) cv. Skyfall in 2015, (**C**) cv. Skyfall in 2016. Coloured symbols indicate a significant POD_0_SPEC–physiology association; black symbols indicate no statistically significant physiological response. Grey regions on plots indicate the period following an observed significant effect of O_3_ on flag leaf CCI.

**Figure 5 plants-08-00084-f005:**
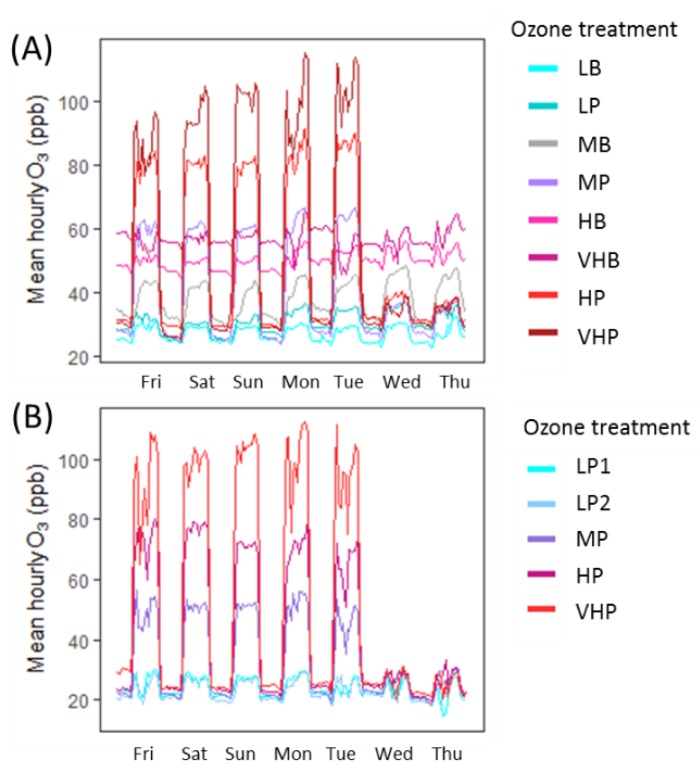
Average hourly O_3_ exposure concentrations in (**A**) 2015 and (**B**) 2016. Values are shown for a one-week period, averaged over the entirety of each growing season. Each treatment has been categorised based on the 24-hour mean exposure (Low, Medium, High, Very High) and the characteristic profile of exposure (peak or background). Treatments were applied five days out of seven to mimic real-world O_3_ exposure.

**Figure 6 plants-08-00084-f006:**
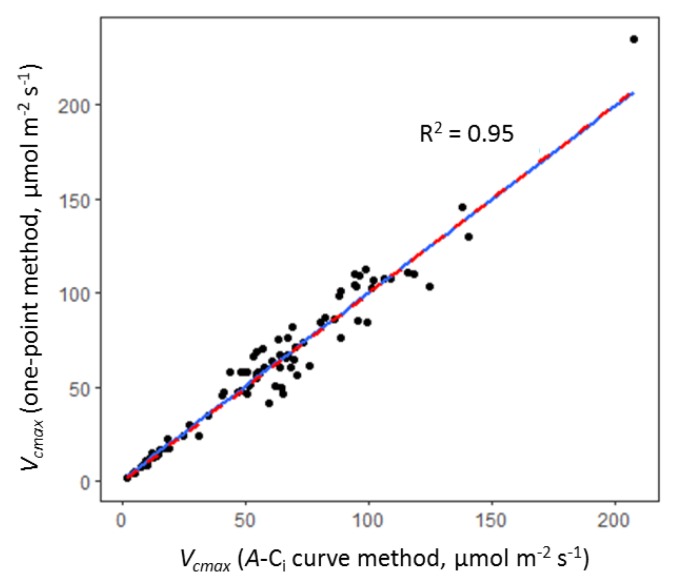
Plot of *V_cmax_* values derived from *A*–C_i_ curves versus *V_cmax_* values calculated using the one-point method [75] from the corresponding *A_sat_* value extracted from each curve (A at 400 µmol mol^−1^ CO_2_). The blue line represents the linear regression model fit (*p* < 0.001, adjusted *r^2^* = 0.95; line equation, y = 0.99x + 1.33). The red dashed line represents the line of x = y. Data for this comparison comprise a subset of the *A*–C_i_ curve dataset used in this study.

**Figure 7 plants-08-00084-f007:**
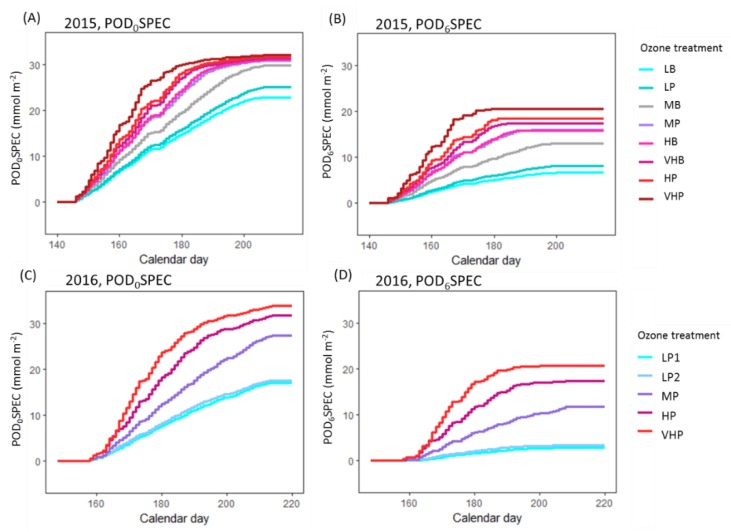
Modelled O_3_ flux over time in the different O_3_ treatments. (**A**) POD_0_SPEC in 2015, (**B**) POD_6_SPEC in 2015, (**C**) POD_0_SPEC in 2016, (**D**) POD_6_SPEC in 2016. Each O_3_ treatment in both years was categorised based on the 24-hour mean exposure (L = low, M = medium, H = high, VH = very high) and the characteristic profile of exposure (P = peak, B = background).

**Figure 8 plants-08-00084-f008:**
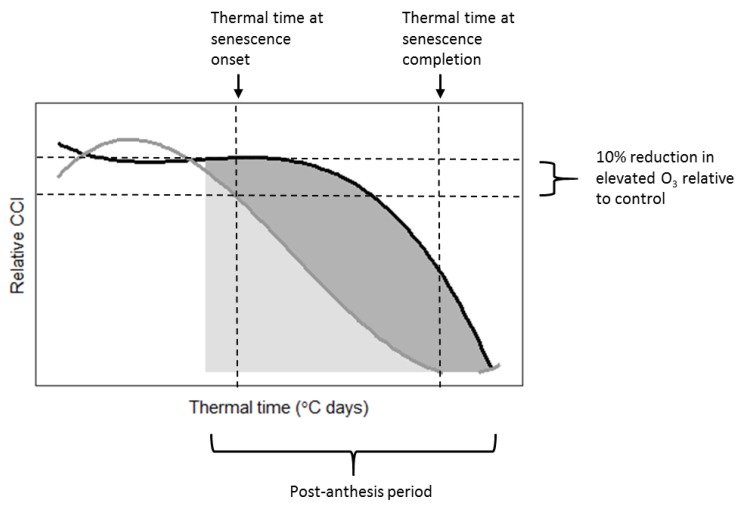
Summary of methods used to derive: i) Thermal time at leaf senescence onset, defined as a 10% reduction in relative CCI in the elevated treatment (grey line) relative to the control (black line); ii) thermal time at senescence completion, defined by the x-abscissa of the treatment regression line; iii) the post-anthesis integral of the regression curve, indicated on the plot as shaded regions (Post-anthesis period in 2015 = 1142 °C days onwards). Diagram not to scale.

**Table 1 plants-08-00084-t001:** Summary of ozone (O_3_) treatments administered in the 2015 and 2016 experiments. 24-h mean, AOT40, and mean daily peak O_3_ have been calculated over the full O_3_ exposure period, whereas the mean POD_0_SPEC, POD_0_SPEC, and POD_6_SPEC quantifies exposure in the flag leaf only (i.e., calculated over the period following flag leaf emergence).

Season	Ozone Treatment	24-h Mean (ppb)	AOT40 (ppm h)	Mean Daily Peak O_3_ (ppb)**	Mean Daily POD_0_SPEC (mmol m^−2^ PLA day^−1^)	POD_0_SPEC (mmol m^−2^ PLA)	POD_6_SPEC (mmol m^−2^ PLA)
2015	LB	26.94	0.002	33.21	0.43	22.87	6.64
	LP	30.39	0.02	36.44	0.46	25.19	8.17
	MB	37.42	4.19	47.74	0.57	29.91	13.03
	MP	40.39	14.51	67.59	0.69	30.99	15.95
	HB	50.06	12.49	56.73	0.71	31.10	15.8
	HP	50.14	28.56	91.90	0.82	31.79	18.48
	VHB	56.81	19.45	66.28	0.78	31.42	17.36
	VHP	55.73	40.03	116.55	1.07	32.16	20.55
2016	LP1	23.42	0.01	31.44	0.36	17.66	3.47
	LP2	22.05	0.03	30.73	0.34	17.11	2.93
	MP	30.41	6.003	55.75	0.54	27.36	11.90
	HP	39.72	21.25	81.04	0.78	31.87	17.39
	VHP	50.14	37.54	113.93	1.04	33.91	20.72

**Table 2 plants-08-00084-t002:** Accumulated flux (POD_Y_SPEC) at the onset of O_3_-induced senescence for 2015 treatments which exhibited significant accelerated senescence.

Cultivar	O_3_ Treatment (2015)	POD_0_SPEC at Senescence Onset (mmol m^−2^)	POD_6_SPEC at Senescence Onset (mmol m^−2^)
Skyfall	LP	17.8	6.5
	MB	15.3	7.9
	MP	22.0	12.9
	HB	24.7	14.1
	HP	25.1	16.2
	VHB	22.9	14.4
	VHP	25.7	18.6
Mulika	VHP	30.1	20.6

**Table 3 plants-08-00084-t003:** Summary of linear mixed model regression analysis to determine whether accumulated POD_0_SPEC, accumulated POD_6_SPEC, 24-h mean, or AOT40 represent the best predictor of physiology in the 5^th^ thermal time-bin. The lowest AIC (Akaike Information Criterion) for each parameter and cultivar, indicating the best model, is highlighted in grey. The outcome of model selection to determine if the profile of O_3_ exposure (i.e., peak vs. background) was important in the flux-based models is also shown.

Parameter	Cultivar	AIC: POD_0_SPEC	AIC: POD_6_SPEC	AIC: AOT40	AIC: 24-h Mean	O_3_ Profile Important in POD_0_SPEC Model?	O_3_ Profile Important in POD_6_SPEC Model?
CCI	Mulika	−62.8	−60.5	−59.2	−58.3	No	No
	Skyfall	3.7	10.4	12.3	15.0	No	No
*A_sat_*	Mulika	−0.5	1.3	4.1	3.2	No	No
	Skyfall	−82.9	−68.0	−48.1	−54.2	No	No
*V_cmax_*	Mulika	6.1	6.9	9.9	9.0	No	No
	Skyfall	−63.0	−62.9	−34.6	−32.3	No	No
*J_max_*	Mulika	−1.7	0.2	3.2	2.3	No	No
	Skyfall	6.3	7.3	9.2	9.5	Yes	No
*g_sto_*	Mulika	13.4	14.8	17.3	16.5	No	No
	Skyfall	−19.1	−7.3	2.7	−1.6	No	No

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
