# Peer review of "New Insights into Leaf Physiological Responses to Ozone for Use in Crop Modelling"

_plants, 2019, doi:10.3390/plants8040084_

Round 1

Reviewer 1 Report

The paper describes the effect of O3 exposure in two wheat cultivars during two consecutive years. I find interesting the cultivar specificity in the O3 response and how O3 in the short time showed some positive effect in photosynthetic parameters. Moreover, it can be inferred that the acceleration of senescence should be considered as an important determinant for crop yield losses in agricultural system. This is in contradiction with the predominant accepted role for ozone as inactivator of Rubisco, that leads to photosynthesis reduction.

However, I find the manuscript too specific and sometimes difficult to follow, and perhaps some attention can be paid in this aspect in order to make the paper more accessible. I would find convenient a more deep explanation of POD0SPEC and POD6SPEC, either in the introduction or in the results sections.

It is not clear to me why the authors reduced the experiment in 2016 to only one cultivar, and one ozone treatment (peak). Perhaps , this could be explained in the text.

Since one cultivar is more resistant to O3, is this associated with differences in yield?

Figure 1 is to dense, consider to separate in two panels the background and peak figures.

Author Response

Response to reviewer 1

We would like to thank the reviewer for their constructive and useful suggestions.

Reviewer 1:

The paper describes the effect of O3 exposure in two wheat cultivars during two consecutive years. I find interesting the cultivar specificity in the O3 response and how O3 in the short time showed some positive effect in photosynthetic parameters. Moreover, it can be inferred that the acceleration of senescence should be considered as an important determinant for crop yield losses in agricultural system. This is in contradiction with the predominant accepted role for ozone as inactivator of Rubisco, that leads to photosynthesis reduction.

However, I find the manuscript too specific and sometimes difficult to follow, and perhaps some attention can be paid in this aspect in order to make the paper more accessible. I would find convenient a more deep explanation of POD0SPEC and POD6SPEC, either in the introduction or in the results sections.

• In response to this comment, we have added some text to the beginning of the results section at lines 159-168. The new text provides a more comprehensive description of the POD0SPEC and POD6SPEC, and the rationale behind calculating each one.

It is not clear to me why the authors reduced the experiment in 2016 to only one cultivar, and one ozone treatment (peak). Perhaps , this could be explained in the text.

• The study combines data from two independent experiments. The first compared physiological response across two wheat cultivars. In the second, the experiment focussed on capturing the physiological response of the more ozone-sensitive cultivar – Skyfall – and it was also decided that fewer treatments were needed.  As both experiments shared very similar experimental design, and the same physiological measurement protocols were applied, data was subsequently pooled to produce a more robust dataset for analysis. We have added some text to the methodology section at lines 500-505 to explain that the differences between the experiments relate to them being independent experiments before the data was subsequently pooled.

Since one cultivar is more resistant to O3, is this associated with differences in yield?

• The physiological response of wheat to O3 exposure – in particular, the response of photosynthesis and leaf senescence to exposure – is the principal focus of this paper. We have therefore not included yield responses in our results. However, analysis of the yield response of Skyfall to O3 exposure found that it is relatively O3-sensitive compared to other cultivars tested (Harmens et al 2018, Atmospheric Environment, 173, 1-5), which ties in with the higher sensitivity in terms of physiological response observed in Skyfall in this study.

Figure 1 is to dense, consider to separate in two panels the background and peak figures.

• We take a different view with regards to Figure 1 than that of the reviewer. Care has been taken when constructing the figure to ensure that the different treatment lines can be distinguished, using colour-coding as well as different line-types. Having all of the O3 treatments on the figure also allows the reader to directly compare the response in peak and background treatments – something that would not be possible if peak and background treatments were separated out into different panels. We have therefore not made any changes to this figure.

Reviewer 2 Report

The manuscript is well written and describes an interesting experimental study with implications to ozone effect modelling. The drawback that it only deals with one species and two cultivars which doesn’t really indicate how large the variability actually might be is compensating by highlighting the mechanistic link between concentration, uptake and damage. The paper is also quite well balanced and holds a quite good overview about the recent literature. The only major difficulty I have is a technical one, namely that the materials and methods section is put at the end. I don’t know if this is in line with the journals rules and I see that it has some advantages to lead the reader right into results and discussion without going to the laborious task of reading the methods first. However, the current structure leads to redundant descriptions of necessary methodological features in the introduction and result session that is quite irritating. I therefore would suggest implementing a somewhat more traditional structure and get rid of the redundancies.

Apart from that, I only have some minor things to comment.

L10: ‘Forecasting’ is a big word that is almost always exaggerated because any estimates for the future are based on environmental and management scenarios. Therefore, ‘scenario analysis’ is usually the term of choice.

L63: What about Ghude et al. in this list of examples?

L70ff: An overview about the way ozone damages the leaf should indicate defense systems that are inherent in practical any plant – and that may lead to carbon costs even before any damage is actually observed. This may also help explain, why threshold modelling does not (always) perform best regarding the flux based damage estimates. See e.g. Tiwari et al. and Jolivet et al. for more explanations

L84ff: I appreciate that mini-review about model approaches but would also mention investigations that are done at other species or not species-specifically since the important thing is the model development as such. With this regard, I think that Reich 1987 is the earliest one that suggests a linear growth reduction in response to ozone. An extreme in mechanistic representation of ozone damages is described in Ploechl et al. and an interesting approach of medium complexity that also accounts for increased respiration losses has been presented by Deckmyn et al.

L123ff: Skip this paragraph, conclusions should not be presented in the introduction.

L393ff: The effect of positive responses of physiological processes to low ozone concentrations is not new in the literature (as is already cautiously indicated) and is increasingly discussed recently. Please consider using the term ‘hormesis’ for this phenomenon. A current view of this process is presented in Agathokleous et al.

L415ff: You might have a look beyond crops and indicate that some similar results have been found on various trees (e.g. Hu et al, Xu et al.).

Indicated references:

Agathokleous, E., Belz, R.G., Calatayud, V., De Marco, A., Hoshika, Y., Kitao, M., Saitanis, C.J., Sicard, P., Paoletti, E., Calabrese, E.J., 2019. Predicting the effect of ozone on vegetation via linear non-threshold (LNT), threshold and hormetic dose-response models. Sci. Total Environ. 649, 61-74.

Deckmyn, G., Op de Beeck, M., Löw, M., Then, C., Verbeeck, H., Wipfler, P., Ceulemans, R., 2007. Modelling ozone effects on adult beech trees through simulation of defence, damage, and repair costs: Implementation of the CASIROZ ozone model in the ANAFORE forest model. Plant Biol. 9, 320-330.

Ghude, S.D., Jena, C.K., Kumar, R., Kulkarni, S.H., Chate, D.M., 2016. Impact of emission mitigation on ozone-induced wheat and rice damage in India. Current Science 110 1452-1458.

Hu, E., Gao, F., Xin, Y., Jia, H., Li, K., Hu, J., Feng, Z., 2015. Concentration- and flux-based ozone dose–response relationships for five poplar clones grown in North China. Environ. Pollut. 207, 21-30.

Jolivet, Y., Bagard, M., Cabané, M., Vaultier, M.-N., Gandin, A., Afif, D., Dizengremel, P., Le Thiec, D., 2016. Deciphering the ozone-induced changes in cellular processes: a prerequisite for ozone risk assessment at the tree and forest levels. Ann. For. Sci. 73, 923-943.

Plöchl, M., Lyons, T., Ollerenshaw, J., Barnes, J., 2000. Simulating ozone detoxification in the leaf apoplast through the direct reaction with ascorbate. Planta 210, 454-467.

Reich, P.B., 1987. Quantifying plant response to ozone: a unifying theory. Tree Physiol. 3, 63-91.

Tiwari, S., Grote, R., Churkina, G., Butler, T., 2016. Ozone damage, detoxification and the role of isoprenoids - new impetus for integrated models. Functional Plant Biology 43, 324-326.

Vlachokostas, C., Nastis, S.A., Achillas, C., Kalogeropoulos, K., Karmiris, I., Moussiopoulos, N., Chourdakis, E., Banias, G., Limperi, N., 2010. Economic damages of ozone air pollution to crops using combined air quality and GIS modelling. Atmos. Environ. 44, 3352-3361.

Xu, Y., Shang, B., Yuan, X., Feng, Z., Calatayud, V., 2018. Relationships of CO2 assimilation rates with exposure- and flux-based O3 metrics in three urban tree species. Sci. Total Environ. 613, 233-239.

Author Response

Response to reviewer 2

We would like to thank the reviewer for their useful and constructive comments and suggestions.

Reviewer 2:

The manuscript is well written and describes an interesting experimental study with implications to ozone effect modelling. The drawback that it only deals with one species and two cultivars which doesn’t really indicate how large the variability actually might be is compensating by highlighting the mechanistic link between concentration, uptake and damage. The paper is also quite well balanced and holds a quite good overview about the recent literature. The only major difficulty I have is a technical one, namely that the materials and methods section is put at the end. I don’t know if this is in line with the journals rules and I see that it has some advantages to lead the reader right into results and discussion without going to the laborious task of reading the methods first. However, the current structure leads to redundant descriptions of necessary methodological features in the introduction and result session that is quite irritating. I therefore would suggest implementing a somewhat more traditional structure and get rid of the redundancies.

• We understand the reviewers’ concerns relating to the inclusion of methodological information in the results section. The positioning of the materials & methods section at the end of the manuscript is in line with the author guidelines and manuscript template for Plants, and this meant that it was necessary to include basic information about the O3 treatments at the beginning of the results section. While we haven’t changed the manuscript structure so that we remain aligned with the author instructions, we have removed some text from the methods section that was unnecessarily repeating treatment descriptions at the beginning of the results section. 

Apart from that, I only have some minor things to comment.

L10: ‘Forecasting’ is a big word that is almost always exaggerated because any estimates for the future are based on environmental and management scenarios. Therefore, ‘scenario analysis’ is usually the term of choice.

• In response to this comment we have amended the wording in this first sentence. It now reads ‘Estimating food production under future air pollution and climate conditions in scenario analysis depends on accurately modelling ozone (O3) effects on yield’ at lines 10-11. 

L63: What about Ghude et al. in this list of examples?

• The study by Ghude et al applies concentration-response functions alongside modelled surface O3 concentrations to estimate O3-induced yield loss. The sentence at line 63 refers to studies that have integrated ozone effect functions based on O3 flux into crop model formulations, allowing the impact of O3 on physiology and yield to be calculated dynamically at hourly or daily time-steps. While we don’t think that line 63 is the right place to include this reference, the reviewer is right to highlight that the Ghude et al study had been omitted from our summary of the literature, and we have added it as a reference at line 42.

L70ff: An overview about the way ozone damages the leaf should indicate defense systems that are inherent in practical any plant – and that may lead to carbon costs even before any damage is actually observed. This may also help explain, why threshold modelling does not (always) perform best regarding the flux based damage estimates. See e.g. Tiwari et al. and Jolivet et al. for more explanations

• In response to this comment, we have added some additional text at lines 73-77. The new text explains that ROS form in the apoplast following O3 uptake, and plants have in-built mechanisms to detoxify ROS, albeit at a carbon cost that may result in a loss of productivity before visible symptoms appear. The studies by Tiwari et al and Jolivet et al have been added as references in this paper.

L84ff: I appreciate that mini-review about model approaches but would also mention investigations that are done at other species or not species-specifically since the important thing is the model development as such. With this regard, I think that Reich 1987 is the earliest one that suggests a linear growth reduction in response to ozone. An extreme in mechanistic representation of ozone damages is described in Ploechl et al. and an interesting approach of medium complexity that also accounts for increased respiration losses has been presented by Deckmyn et al.

• We thank the reviewer for bringing these studies to our attention. We have now brought the papers by Reich (1987) and Deckmyn et al (2007) into our review of proposed methods for modelling O3 effects on photosynthesis and senescence at lines 90-93 and 97-99. We have added the paper by Plöchl et al to the Discussion at lines 450-453, where we discuss the limitations of thresholds for O3 accumulation in models, and suggest a mechanistic representation of O3 detoxification could lead to improved accuracy.

L123ff: Skip this paragraph, conclusions should not be presented in the introduction.

• We recognise that including a short summary of conclusions at the end of the introduction is unusual in a research paper. However, it is in line with the author instructions and template for Plants. The plants author guidelines give the following instructions for the Introduction section: “Finally, briefly mention the main aim of the work and highlight the main conclusions”. We have therefore not made any changes to the text here, to remain aligned with the author guidelines.

L393ff: The effect of positive responses of physiological processes to low ozone concentrations is not new in the literature (as is already cautiously indicated) and is increasingly discussed recently. Please consider using the term ‘hormesis’ for this phenomenon. A current view of this process is presented in Agathokleous et al.

• Many thanks for bringing the paper by Agathokleous et al to our attention. We have added some text at lines 416-418 explaining that the stimulation of growth at low doses of a toxin is known as hormesis. We have also made an addition to lines 428-430 suggesting that a bi-phase hermetic dose-response relationships such as described by Agathokleous et al (2019) could be considered in future ozone effect model functions.

L415ff: You might have a look beyond crops and indicate that some similar results have been found on various trees (e.g. Hu et al, Xu et al.).

• In response to this comment we have added some text to this paragraph at lines 437-438, indicating that flux has also been shown to be better than concentration-based metrics at predicting poplar biomass and assimilation rate in urban trees, under O3 exposure. Both of the suggested references have been added to the text.
